# ECO grad: Error Correcting Optimization for Quasi-Gradients, a Variable Metric DFO Strategy

## Abstract

We introduce a *Quasi-Gradient* method using 0th order directional derivatives and quasi-Newton like updates. Empirically, our method reduces $d$-dependence of zeroth-order problems to an effective $\approx d \cdot m$ factor $1/d \leq m \leq 1$, with only a small linear increase in compute. We show this holds under Lipschitz bounds and on practical tasks. While compressive sensing achieves similar gains with sparse gradients, our approach applies to any gradient geometry. It exploits high cosine similarity and stable gradient norms along neighboring steps, ultimately requiring fewer samples to correct the estimator. Applications include policy optimization, model-free reinforcement learning, function smoothing, evolutionary methods, efficient JVPs (e.g. in JAX), learning from simulation, and related areas. We include a probing framework that leverages convergence bounds to detect when a gradient estimator is no longer aligned with new samples, helping prevent non-descent steps. We also introduce the *ECO estimator* a least-change secant update that results in a specific LMS adaption, which achieves $O(e^{-k/d})$ convergence in gradient MSE, while Monte Carlo averaging is sub-exponential $O(\frac{d+1}{d+k+1})$. Finally we provide performance results comparing directional SGD to quasi-GD, alone and with adaptive optimizers. As models grow, our approach bridges the gap between full-gradient methods and large scale derivative free optimization. We hope to motivate further research in quasi-gradient techniques for simulation and exploratory learning.

## 1 0th-Order Gradient Estimation

## 2 Directional Derivatives and Gradient Estimators

### 2.1 Directional Derivatives

In the standard literature a *Directional Derivative* is defined as $(\nabla f(\boldsymbol{x}) \cdot \boldsymbol{u})\boldsymbol{u}$ or $\langle \nabla f(\boldsymbol{x}), \boldsymbol{u} \rangle \boldsymbol{u}$, Nesterov & Spokoiny (2017), we refer to it as $(v)\boldsymbol{u}$ for convenience and because $v = \langle \nabla f(\boldsymbol{x}), \boldsymbol{u} \rangle$ will be utilized separately. It is also known as a *Forward Gradient* and a forward *Jacobean Vector Product* Baydin et al. (2022). What remains ambiguous, and we find important to address is defining $\boldsymbol{u}$. The most common form is $\boldsymbol{u} \sim \mathcal{N}(\boldsymbol{0}, \boldsymbol{I})$ this satisfies the definition of Gaussian Smoothing: $\mathbb{E}\left[(v)\boldsymbol{u}\right] = \nabla f(\boldsymbol{x})$, and can be implemented directly as a form of SGD Nesterov & Spokoiny (2017). The single necessary assumption of *smoothing* is unbiasedness, but there isn't a specification for variance or distribution of $\boldsymbol{u}$. Unbiasedness does not mean the distribution of $\boldsymbol{u}$ can't be biased e.g. Rademacher or Bernoulli, de-biasing can also be performed after sampling Ye et al. (2019). However this can all lead to potentially harmful asymptotics that slow SGD, we continue this discussion here A.1.

Smoothing error is generally measured by MSE, but we believe this doesn't capture enough perspective on gradient estimators. Our approach focuses on cosine similarity and norm separately. MSE can be viewed as capturing two dimensions of error: 1) How large is the angle between the estimator and true gradient? 2) How large is the difference between norms of the estimator and gradient? Between the two, closer angle (larger cosine) is most expensive and important to estimate

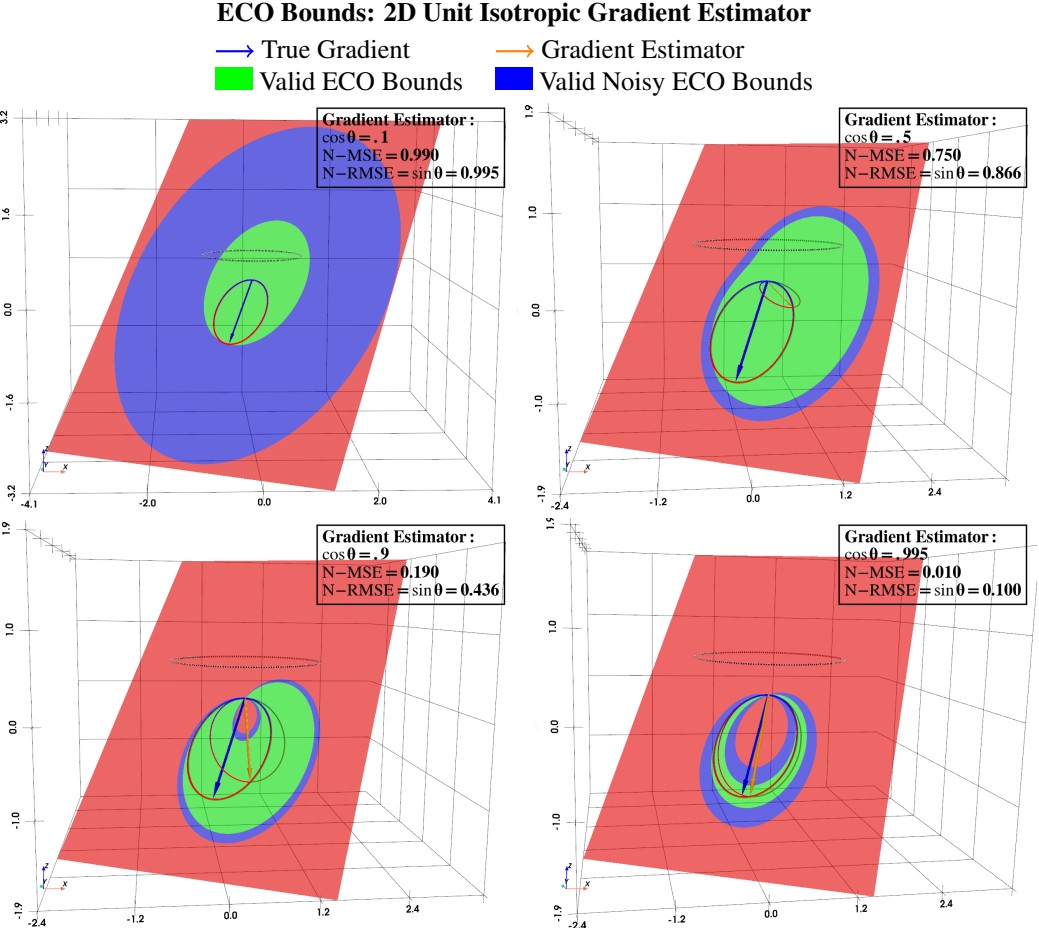

Figure 1: Estimators that satisfy lemma 2.1 and corollary 3.1. By calculating the accuracy as a convergence expectation, typically as $\cos\theta$, we can draw precise bounds around where our *Directional Derivatives* can appear relative to our estimate so that they are still feasible on the ring of the true gradient. We see that as the accuracy improves, the feasible region converges to the true gradient ring. If a Directional Derivative landed outside of these bounds at any point, we would know that the true gradient has changed. Plotting code (TBE). The procedure is covered in section 3.

accurately. This is because by selecting the correct distribution, we can calculate the MSE optimal estimate (that determines the norm) for any given $\cos\theta$. Define the uniform unit sphere distribution as $\boldsymbol{u} \sim \mathrm{Unif}(\mathcal{S}_{d-1})$ where $\boldsymbol{u} = \boldsymbol{v}/\|\boldsymbol{v}\|$, $\boldsymbol{v} \sim \mathcal{N}(\boldsymbol{0}, \boldsymbol{I})$. We believe $\boldsymbol{u}$ exhibits a very ideal theoretical perspective as compared to other distributions, because it satisfies what we call a *True Directional Derivative* lemma 2.1. It's also the only fully independent, identically distributed, and uniform variable on $\mathcal{S}_{d-1}$.

**Lemma 2.1.** *Define $f(\boldsymbol{x})$ such that $\nabla f(\boldsymbol{x})$ is continuous, and let $\boldsymbol{u} \in \mathbb{R}^d$, s.t. $\|\boldsymbol{u}\| = 1$. Then with $\theta = \angle(\nabla f, (v)\boldsymbol{u})$*

$$\frac{\|\nabla f(\boldsymbol{x}) - (v)\boldsymbol{u}\|}{\|\nabla f(\boldsymbol{x})\|} = \sin\theta, \qquad \frac{\|(v)\boldsymbol{u}\|}{\|\nabla f(\boldsymbol{x})\|} = \frac{|v|}{\|\nabla f(\boldsymbol{x})\|} = \cos\theta, \qquad (2.1)$$

And we have $0 \le \sin\theta \le 1$, and $0 \le \cos\theta \le 1$ [proof B.1]. These relationships wouldn't exist without $\|\boldsymbol{u}\| = 1$, and they are the key insight behind how we predict if a gradient estimator ((2.2) or (2.3)) is accurate without having access to $\nabla f(\boldsymbol{x})$. By the Pythagorean theorem, we know a directional sample that obtains the smallest possible MSE for a given positive $\cos\theta$ will be a true directional derivative. This why we use $\mathrm{Unif}(\mathcal{S}_{d-1})$ for the methods below, we try to replicate this symmetry as closely as possible. We also now have $\mathbb{E}[(v)\boldsymbol{u}] = d^{-1}\nabla f(\boldsymbol{x})$ [B.3].

For reference $\lim_{d\to\infty} \mathrm{Unif}(\mathcal{S}_{d-1}) \sim \mathcal{N}(\boldsymbol{0}, \frac{1}{d}\boldsymbol{I})$ has a rate of $O(d^{-1/2})$ [B.6]. When $d$ is large we may even sample directly from $\mathcal{N}(\boldsymbol{0}, \frac{1}{d}\boldsymbol{I})$ without an expensive normalization. A brief discussion

on finite differences to approximate directional derivatives with high accuracy and exotic estimators can be found here A.2. For clarity in further sections we say the gradient normalize root mean square error of our directional derivative: *N-RMSE* = $\sin \theta$. We also use $\nabla f(\boldsymbol{x})$ and $\boldsymbol{g}$ interchangeably.

## 2.2 GRADIENT ESTIMATORS

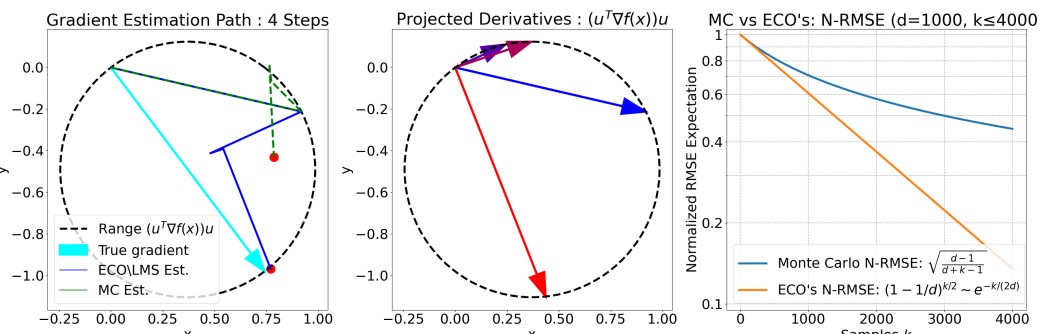

Figure 2: Left plot: path of a Monte Carlo gradient estimator normalized to be MSE minimizing, and ECO's method. Center plot: directional derivatives used by the estimators. Right plot: derived convergence rates of Normalized RMSE.

When a large parameter count makes other methods difficult, *Monte Carlo Averaging* is a well known method for estimating the gradient. Even outside of 0th order optimization, it is the primary method used in batched stochastic sampling; this encompasses alternative gradient estimators such as policy and natural gradients in RL. We define it for our specific setting as:

$$\tilde{\boldsymbol{g}} = \frac{d}{N} \sum_{k=1}^{N} (v_k)\boldsymbol{u}_k, \qquad \begin{array}{l} N \text{ - Sample size.} \\ d \text{ - Problem dimensions.} \end{array} \tag{2.2}$$

It's commonly understood that Monte Carlo estimation converges to the population mean in $O(k^{-1/2})$, this is true for the RMSE: $\|\tilde{\boldsymbol{g}} - \boldsymbol{g}\|$ of our estimator. While being an unbiased estimator Duchi et al. (2014), it doesn't produce the (approximately) smallest possible MSE/RMSE for $k$ samples in expectation. But fortunately Gao & Sener (2022) has already solved this for Gaussian admitting $\frac{k}{d+k+1}$, our $S_{d-1}$ result is a bit different : $\frac{k}{d+k-1}\tilde{\boldsymbol{g}}_k = \hat{\boldsymbol{g}}_k$. With $\|\boldsymbol{u}\| = 1$ and $d$ multiplier, it's evident that $\|\hat{\boldsymbol{g}} - \boldsymbol{g}\|/\|\boldsymbol{g}\| \leq 1$, with $O((\frac{d-1}{d+k-1})^{1/2})$ convergence, [proof B.3].

Now we introduce *ECO's Method*. (We have renamed this method temporarily to hide an author's identity.) This is our new application of established methods that achieves [proof B.4] exponential $O((1 - \frac{1}{d})^{k/2})$ N-RMSE convergence, figure 2. There are many ways to interpret and arrive at this update. To honor Quasi-Newton methods, we define the secant constraint and variable metric for Langrange form.

**ECO's Method** [Proof B.2] *Solve* $\min_{\hat{\boldsymbol{g}}_k} \|\hat{\boldsymbol{g}}_k - \hat{\boldsymbol{g}}_{k-1}\|^2$ *s.t.* $\langle \hat{\boldsymbol{g}}_k, \boldsymbol{u} \rangle = v$. *Admits:*

$$\boxed{\hat{\boldsymbol{g}}_k = \hat{\boldsymbol{g}}_{k-1} + \frac{(v - \hat{\boldsymbol{g}}_{k-1}^T\boldsymbol{u})\boldsymbol{u}}{\boldsymbol{u}^T\boldsymbol{u}}}, \;\; \textit{iff} \;\; \|\boldsymbol{u}\| = 1. \; \rightarrow \; \boxed{\hat{\boldsymbol{g}}_k = \hat{\boldsymbol{g}}_{k-1} + (v - \hat{\boldsymbol{g}}_{k-1}^T\boldsymbol{u})\boldsymbol{u}} \tag{2.3}$$

It is already a MSE minimizing estimator, and N-RMSE $\leq 1$ almost certainly by the results of lemma 2.1. To attribute the recurrence we may also call it the *Least Change Gradient Estimator* in a Euclidean sense, equivalent to Broyden's Method but for gradients instead of the Hessian. It's identical to the *N-LMS Update* and uniformly random *Kaczmarz Update* Gower & Richtárik (2015) with a known optimal learning rate $l = 1$. Going forward we will use these names interchangeably. For intuition on why ECO's Method is exponential even though MC and LMS have the same $O(d \cdot k)$ operational dependence, see discussion A.3 where we also mention *Block ECO's Method* and Orthogonal directions. To see how ECO's Method and MC perform on a static gradient: figure 4.

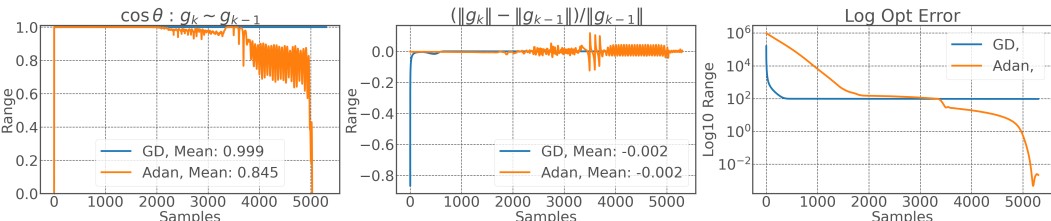

Figure 3: $d = 100$ Rosenbrock. Lipschitz $L$ at $x_0$. LR for GD $= 6/L$, Adan $= 300/L$ [source].

## 3 ECOGRAD AND PROBING FRAMEWORK

The *ECOgrad* framework is what we call a Quasi-Gradient method as it improves a gradient estimate at parameter vector $x_k$ that was already established at previous $x_{k-n}$'s. We are motivated by the empirical observation that full gradients remain significantly correlated between descent steps figure 3, even with aggressive learning rates. In Nesterov's analysis of DFO (source) and in other works (sources), we see $O(d/\epsilon)$ or improved bounds with $d$-dependence. Under Lipschitz optimal, $n = d/\epsilon$ is oracles to achieve $\epsilon$ bounds on the stationary point. With $\mathbb{E}[\cos \theta^2] = d^{-1}$ B.3, let $d = 100$ gives us $\mathbb{E}[\cos \theta^2] = .01$, if our method averages $\cos \theta^2 = .1$ without additional queries (this is like $\approx 10$ oracle calls for free) we may need only $m \approx 1/10$ of our original $n$ queries to achieve $\epsilon$ margin. We might call $md$ the effective dimension size.

To achieve this effect, one strategy would be to naively update the estimator, but this could fall short. Monte Carlo estimation is dependent on $k$; as samples are received each has less impact on the estimate leading to stagnation. However *ECO's Method* has exponential convergence and will adapt in time. This effect alone is enough to produce a variance reducing strategy as seen in *SEGA*, that generalizes to eliminate distribution bias and even operate on sub-spaces Hanzely et al. (2018). Yet estimator convergence is still a stationary assumption and depends on the dimension size, it follows that a step must be smaller to allow the method to adapt quickly enough to changes in the gradient. The method may not beat the $d$-dependent (optimization) lower bound of the underlying sample strategy. When analyzing impacts of large step size or non-characteristic surfaces we find situations that form bad estimates, like non-descent directions leading to oscillation and asymmetrically worsened progress. The next gradient may form a greater than $90 \deg$ angle with the previous, or the norm might reduce significantly (common for stationary points). In both cases, resetting the estimator to zero or reducing it's size would result in a faster update to the true gradient and even guarantee a descent direction. Reset and shrinkage has found success already in 2nd order methods [Ca et al. (2020), Indrapriyadarsini et al. (2020)] and is the basis of our strategy.

Seeking a corrective method that works generally, we developed a system that only depends on the gradient estimator and directional derivatives. Empirically we find that avoiding non-descent directions is especially consequential, our strategy aims to preserve *estimator* descent first and then improve MSE or $\cos \theta$ with the same sampling rate. The added benefit is that we can work with other estimators, like Monte Carlo that is convergent under noise.

### 3.1 BOUNDS AND ECO RATIO

We first begin with the gradient, and enforce lemma 2.1: State $c = \left(1 - \frac{1}{d}\right)^{k/2}$ if we use ECO's Method and $c = \left(\frac{d-1}{d+k-1}\right)^{1/2}$ for MSE minimizing Monte Carlo.

**Corollary 3.1.** *Define $\hat{g}$ such that $c = \sin \angle(g, \hat{g})$, and $\|\hat{g}\|/\|g\| = \cos \angle(g, \hat{g})$ i.e. unit isotropic estimator on $\mathcal{S}_{d-1}$. And $\|u\| = 1$ then*

$$\frac{|u^T \hat{g} - v|}{c\|g\|} \leq 1 \tag{3.1}$$

Proof B.5. These are the strongest definite bounds on directional derivatives we could find when the estimator also satisfies unit isotropic. This is what figure 1 green area visualizes, more conservative bounds in the 2D scenario may lead to false positives. In many dimensions this is consistent, yet from B.3 we know $\mathbb{E}[uu^T] = d^{-1}I$. So even when $d \gg 1$ it is still possible for a $(v)u$ to appear up

to a residual $= c\|\boldsymbol{g}\|$, but very unlikely. We rely on the distribution of our samples to state what is *improbable*, not *impossible*.

**Corollary 3.2.** *Proof B.6. Assume corollary 3.1 then*

$$\mathbb{E}\left[|\boldsymbol{u}^T\hat{\boldsymbol{g}} - v|^2\right] = \frac{c^2\|\boldsymbol{g}\|^2}{d} \tag{3.2a}$$

$$\mathbb{E}_\alpha\left[|\boldsymbol{u}^T\hat{\boldsymbol{g}} - v|^2\right] = \frac{c^2\alpha^2\|\boldsymbol{g}\|^2}{d} \tag{3.2b}$$

$\alpha$ is the Gaussian two-sided significance level, found in a CI table or by $\Phi^{-1}$. Next by 3.1 we recognize $\|\boldsymbol{g}\| = \|\hat{\boldsymbol{g}}\|/\cos\angle(\boldsymbol{g},\hat{\boldsymbol{g}}) = \|\hat{\boldsymbol{g}}\|/\sqrt{1-c^2}$. We can even define $\overline{\boldsymbol{g}} = \hat{\boldsymbol{g}}/\sqrt{1-c^2}$ as the *norm error* minimizing estimator.

The **ECO Ratio:**

$$\mathcal{M}(v,\boldsymbol{u},\hat{\boldsymbol{g}},c,\alpha) \models M(v) = \frac{|\boldsymbol{u}^T\hat{\boldsymbol{g}} - v|\sqrt{d(1-c^2)}}{\alpha c\|\hat{\boldsymbol{g}}\|} \tag{3.3}$$

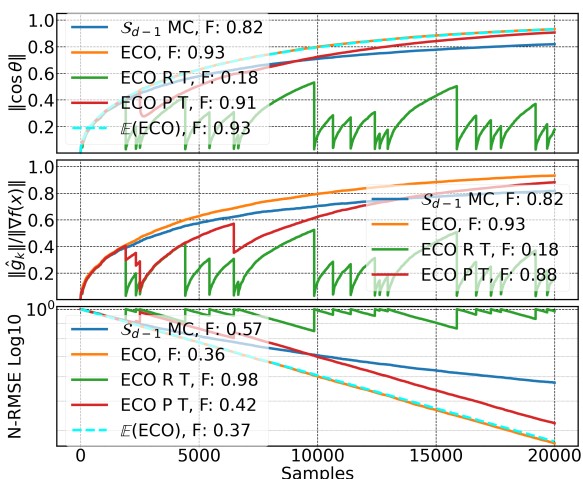

proof B.6 Iff $M(v) > 1$ we fail to support $\hat{\boldsymbol{g}}$ is a continuing estimator of $\boldsymbol{g}$ up to $c$ (our expected estimator N-RMSE (link to appendix as to why we call it that)), we set $\hat{\boldsymbol{g}}_k = (v)\boldsymbol{u}_k$, $c = \sqrt{1-d^{-1}}$, and begin to improve $\hat{\boldsymbol{g}}$ and $c$ again in forward steps. A benefit of this ratio stems from it's independence of $\nabla\boldsymbol{x}$ or the higher moments of $f(\boldsymbol{x})$, allowing it to work unconditionally in many situations. The only requirement is L-smoothness (by $(v)\boldsymbol{u}$ not even by $\nabla f(\boldsymbol{x})$). Under convexity assumptions, parameter count, interactions, and step size, the optimal $\alpha$ may vary but this is beyond the scope of our current work.

Figure 4: $d = 10^5$, $\alpha_{\mathcal{N}} \approx 3.3$. T - Students's t. R - Resets. P - Partial resets. F - Final Value.

We conclude by mentioning certain handwaving, necessary to achieve our result: 1) In Equation (3.2b) we assume $\alpha$ is Gaussian but $\boldsymbol{u} \sim \mathcal{S}_{d-1}$, for small $d$ this appears trivial, at large $d$ we know $\mathrm{Unif}(\mathcal{S}_{d-1}) \sim \mathcal{N}(\boldsymbol{0}, \frac{1}{d}\boldsymbol{I})$. 2) In reality the isotropic assumption of corollary 3.1 is never exact. We hypothesize that when $\mathbb{E}[\boldsymbol{u}\boldsymbol{u}^T] = (d)^{-1}\boldsymbol{I}$, then $\|\hat{\boldsymbol{g}}\|/\|\boldsymbol{g}\|$ is convergent to $\cos\angle(\boldsymbol{g},\hat{\boldsymbol{g}})$ in a approximately t-distributed manner, independent of dimension size. We provide our evidence in discussion A.4. Because calculating $\alpha \sim t(\nu)$ at each step can be expensive, we also provide an accurate polynomial interpolate $\alpha_t(\alpha_{\mathcal{N}}, \nu)$ in our repository (source). The DOF $\nu$ represents steps since last reset.

## 3.2 NOISY ECO RATIO

We begin with $\tilde{v}_k = v_k + e_k$. Assume $e_k \sim \mathcal{N}\left(0, \sigma^2\right)$ so that: $e_k$ is independent of $v_k$. $\mathbb{E}[e_k] = 0$. And $\mathbb{E}[(e_k)^2] = \sigma^2$.

The **Noisy ECO Ratio:**

$$\gamma = \frac{\alpha c\|\hat{\boldsymbol{g}}\|}{\sqrt{d(1-c^2)}}, \quad \tilde{M}(\tilde{v}) = \sqrt{\frac{(\boldsymbol{u}^T\hat{\boldsymbol{g}} - \tilde{v})^2}{\gamma^2 + \sigma^2}} \leq 1 \tag{3.4}$$

Proof B.7. We see that after adding noise, Equation (3.4) simply contributes a static threshold that eventually dominates the boundary. It is in root form for consistency but that is not necessary. If we

are using ECO's Method then a variable learning rate is theoretically optimal, proof B.8. We get the recurrence:

$$\hat{\mu}_k = \frac{c_k^2 \|\hat{\boldsymbol{g}}_k\|^2}{d(1-c_k^2)}, \quad l_k \approx \frac{\hat{\mu}_k}{\hat{\mu}_k + \sigma_e^2}, \quad c_{k+1}^2 = c_k^2 \cdot (1 - l_k d^{-1})$$

$$\hat{\boldsymbol{g}}_{k+1} = \hat{\boldsymbol{g}}_k + l_k(v - \hat{\boldsymbol{g}}_k^T \boldsymbol{u}_k)\boldsymbol{u}_k$$

(3.5)

Note: Updating $c_{k+1}$ is not the same as updating $c_{k+1}^2$ we need to take the root separately if $c_{k+1}$ is needed.

$\sigma_e^2$ can be estimated adaptively or known at first, it may also help to alter it's significance by constant factors. Holding $\sigma^2$ and $\|\hat{\boldsymbol{g}}_k\|$ constant we would find that $\lim_{k\to\infty} c_k \to n$ where $0 < n < 1$ and $l_k$ approaches 0. This is an expected behavior, the LMS filter does not fully converge under noise (source). For moderate noise such as (smoothing) non-smoothness, discontinuous function estimation, and finite difference stencil error, ECO's Method should be viable. When noise becomes a significant portion of the true gradient norm, consider: 1) Averaging multiple directional derivatives, or use an over-determined stencil regression. 2) Switch to Monte Carlo Estimation at such point that LMS progress is estimated to be slower. 3) For finite/semi-finite SGD, batch *along dimensions* and not along environment or dataset sections, avoiding noisy updates all together.

Option 3 is generally unavailable to full gradient methods, but a widely relevant strategy to DFO and ECOgrad. In (ilya and co) show how Gaussian smoothing can be used to achieve similar results to model based methods and policy gradients, while also seeing nearly linear return for additional network resources. In their work parallelization happens over separate gaming environments, then RNG states and directional perturbations $v$ are transferred as scalar values, requiring minimal bandwidth. This exact strategy can still benefit from our framework while enabling new possibilities. We provide further notes regarding networked asynchronous learning and maximizing compute efficiency when calculating estimators, discussion A.6.

## 3.3 ECOGRAD PARTIAL RESETS

During the hypothetical progress of our optimization, the ECO bounds may be violated but with an insignificant tail (just barely). Both statistical anomalies and the true gradient only changes in norm or angle slightly, are possible. To remedy this we introduce a shrinkage method for our existing $\hat{g}_k$ that also relax the ECO bounds. We provide justification for this approach in discussion A.5. We can solve for the partial reset by increasing $c$ and simultaneously shrink $\hat{g}$ to the norm that would be expected for this increase, such that the ECO Ratio is $< 1$. $n$ references new values, and the iteration $k$ is arbitrary. Our *Reset Boundary* equation:

$$R_1(c_n) = \left| \boldsymbol{u}^T \hat{\boldsymbol{g}} \frac{\sqrt{1-c_n^2}}{\sqrt{1-c^2}} - v \right| - \frac{\alpha c_n \|\hat{\boldsymbol{g}}\|}{\sqrt{d(1-c^2)}} = 0$$

(3.6)

We solve for the smallest $c < c_n \le 1$. This has a quadratic symmetric four root solution, an analytic method is provided: Algorithm 1. Afterwards the estimator must be updated:

$$\hat{\boldsymbol{g}}_n = \hat{\boldsymbol{g}} \frac{\sqrt{1-c_n^2}}{\sqrt{1-c^2}}$$

The *Noisy Reset Boundary*:

$$\gamma_n(c_n) = \frac{\alpha c_n \|\hat{\boldsymbol{g}}\|}{\sqrt{d(1-c^2)}}, \quad \tilde{R}(c_n) = (\boldsymbol{u}^T \hat{\boldsymbol{g}} \sqrt{\frac{1-c_n^2}{1-c^2}} - \tilde{v})^2 - \gamma_n^2(c_n) - \sigma^2 = 0$$

(3.7)

This is now a full quartic due to the noise term. We provide a bracketed secant solver in our code base (here) customized to solve this quickly.

Finally if we use the Student's t significance model $\alpha_t(\alpha_{\mathcal{N}}, \nu)$ our boundary solutions lose their polynomial expectations. Fortunately $\alpha_t$ consistently results in $c_n$ solutions that are smaller than those under $\mathcal{N}$ without strange behaviors. In fact t adjusted significance usually results in a solution where $c_n < 1$, even when $c_n = 1$ normally. Empirically $\alpha_t(\alpha_{\mathcal{N}}, \nu)$ tends to improve optimization performance under partial resets. To solve with $\alpha_t(\alpha_{\mathcal{N}}, \nu)$ refer to our secant implement (here), it

will also return variables $c_n$ and $s$. Set $c_k = c_n$ then DOF $\nu = s$ is calculated by the ln equation in our code:

$$\text{ECO's Method} : \nu = \frac{2\ln(c_k)}{\ln\left(1 - \frac{1}{d}\right)}, \quad \text{Monte Carlo} : \nu = \frac{d-1}{c_k^2} - d + 1 \qquad (3.8)$$

Note: If $v$ is small which is when it is relevant. Monte Carlo $v$ is nearly the same as ECO, but avoids the expensive logs.

From here increment $\nu_{k+1} = \nu_k + 1$ until the next reset is triggered. Or with noise every new sample represents diluted information, calculate with Equation (3.8) for each step.

### 3.4 RESULTS SECTION

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

## A  DISCUSSION

### A.1  DIRECTIONAL PERTURBATION AND DISTRIBUTION

If our intent is to estimate a true gradient, de-biasing is necessary and can be expensive. As we assume little about $f(\boldsymbol{x})$, we . There are other asymptotically unbiased distributions, but we opt for the following to derive our framework in an elegant manner.

### A.2  DIRECTIONAL DERIVATIVE STENCILS

*Approximation:* Directional derivatives are cheap to estimate. By selecting a random unbiased direction $\boldsymbol{u}$ we 1) enable our optimization to make progress independent of individual coordinates even before $k \geq d$ where $k$ represents total optimizer steps. 2) Reduce the problem to an objective in 1D and gain access to reduced finite difference stencils. We can even treat $(v)$ as a black box output in use of exotic methods, like de-noising, metrics, or (source). The following are well known estimates for $(v)$ and a lesser known one.

Incomplete section .

But there is also:

We can see that it is possible to obtain a $O(h^4)$ accurate estimate of $\nabla f(\boldsymbol{x})$ in just $4d$ function evaluations. Gradient stencils exist for $O(h^{2n}), n \in \mathbb{Z}^+$. They may not be worth the effort for floating point accuracy; and Nesterov (source) shows good results can be obtained for non-smooth $f(\boldsymbol{x})$ even with $O(h)$ stencils. But they have another use, obtaining greater accuracy when $f(\boldsymbol{x})$ is significantly discontinuous, such as a reward landscape in offline RL or simulations. We conclude this as an area of further research.

that would need significantly more samples to replicate in higher dimensions.

### A.3  BLOCK ECO'S METHOD INTUITION

An intuition is to see the update as a rapidly convergent solver of a linear system for $\nabla f(\boldsymbol{x})$. We could treat this as a direct interpolation of $f(\boldsymbol{x})$ within a proximal area requiring $d+1$ samples (DFO

source); but canceling the value intercept of $f(\boldsymbol{x})$ allows us to abstract to a system of directional derivatives. In fact, we define the *Block ECO's Method*:

$$\hat{\boldsymbol{g}}_k = \hat{\boldsymbol{g}}_{k-n} + (\boldsymbol{v} - \hat{\boldsymbol{g}}_{k-n}^T \boldsymbol{U})\boldsymbol{U}^T(\boldsymbol{U}\boldsymbol{U}^T)^{-1}, \quad \begin{array}{l} \text{Compute Complexity: } O(d(k/n) \cdot n^2) = O(d(k)n) \\ \text{Convergence Rate: } O\left((1 - \frac{n}{d})^{k/2n}\right) \end{array}$$

(A.1)

In this setting $k$ now increments by $n$. If batch size $n$ is $<< d$ which is our expected setting, the block update will have virtually no additional error improvement, but cost an extra $d * n$ per iterate. On the opposite end, it could make more sense to complete the interpolation $d = n$, providing well established 1st order guarantees and aggressive super-linear methods e.g. Quasi-Newton. For this reason we don't further mention the block update in our framework. We briefly mention that it is possible to amortize the least squares solution using orthogonal $\boldsymbol{U}$, but it still has it's own drawbacks (continue here if time).

### A.4 T DISTRIBUTED SIGNIFICANCE

We argue for dimension invariant t-distributed $\alpha$ by noting there are two opposing forces that scale with dimension size. First as $d$ increases the influence of any one $u_i \in \boldsymbol{u}$ decreases, we know already that $\mathbb{E}[(v)\boldsymbol{u}] = \boldsymbol{g}/d$ and $\mathbb{E}[\cos\theta^2] = 1/d$ will have the effect of smoothing/decreasing the variation of $|v|$ and reducing contribution of any one sample in improving the estimate. Second the estimators require more samples to reach the same MSE optimal accuracy as $d$ increases, we see this with the longer lasting leverage of $c/\sqrt{1-c^2}$, balancing this variance reduction effect. Also see the end of proof B.6. (maybe plots).

### A.5 WHY OUR SHRINKAGE

For the Kaczmarz lemma in proof B.4 we see that the right hand side has an $\boldsymbol{x}_0$ initial point, by setting it to 0 we arrive at our intuitive framework. We chose shrinkage instead of re-deriving our bounds and variables, as it requires an analysis from an FTRL perspective instead of OMD, complicating our setting. It also breaks the approximate symmetry of lemma 2.1. We believe shrinkage is more appropriate for gradients anyway, because:

1. During optimization, the only time we can be completely certain that our gradient estimator does have N-RMSE $< 1$ and $\cos\theta > 0$ is when $\hat{\boldsymbol{g}}_k = [0]_d + (v)\boldsymbol{u}$. Therefore shrinkage is possibly a consistent method that makes the next update to satisfy this criteria more likely.

2. Shrinkage of 0 stationary parameters is synonymous with loss of information, especially for online methods like our LMS adaption. Shrinkage is the method exponential moving averages use to 'forget'.

3. We don't have a guarantee that even after re-deriving the Kaczmarz bounds, convergence rate, and optimal learning rate, it will actually adapt quicker to the true gradient from the anchor point. In fact if we assume in a convex setting that $\mathbb{E}[g] = [0]_d$ over the life of the optimization, it may even hinder convergence.

4. Even in a non-convex setting, convergence to a stationary point on a smooth surface implies a decreasing $\|\nabla f(\boldsymbol{x})\|$. So it's reasonable to consider norm shrinkage may place our estimator within a better range of "steepness" as the stationary point is approached. Additionally, by assuming negligible correlation between parameters we can guess that the angle of the gradient may change even less than the norm on average.

### A.6 BIG COMPUTE ECOGRAD STRATEGIES

For infinite set SGD like temporal differencing and path dependence, we suspect longer sections will work better, up to a Pareto front, as is usually the case in comparing offline methods.

Distributed ECOgrad: Let's have separable no-grad environments or datasets. We break them up into minibatches, assign each a gradient estimator. As we take new steps, allocate a certain amount of queries to updating and validating each gradient estimator with their respective N-RMSE expectation and ECO Ratio. We receive any full or partial reset requests, then depending on the new expected N-RMSE's we allocate a proportional amount of queries to each estimator so that all estimators/minibatches reach a certain tolerable error expectation. This way we can obtain an accurate

net gradient estimate with even less compute (no wasting evaluations on estimators that already have tolerable expectation, and allocate more concurrent compute to those that need it), the tradeoff is more memory. And we don't know what the lower bound of savings are.

Async Batched ECOgrad: Each environment continuously samples directional queries, after a $v$ is received, they are sent to a warehouse on network. The warehouse has lots of memory and high bandwidth, updating the gradient estimators of each environment as they are received according to ECOgrad. We have decoupled how good our estimate will be from how many queries we take at each round, so it should matter less if certain environments have more queries than others. After a certain criteria is hit, such as net expected gradient MSE, the estimates are combined. Possibly equal weighting, or weighted by expected gradient accuracy, or another scheme, and a step is taken. The only high-bandwidth need is transferring new parameters to each environment. However the environments never have to halt sending directional scalars, as even samples from stale but nearby parameters can improve the gradient estimates. We believe this to be an option for massively scaled real-time model free RL, such as a global network of models/agents that adapt to world environments collectively.

ECOgrad SAGA: SAGA but replace the exact gradient calculations with estimators. Or a jacobian approximation.

## A.7 AREAS OF FUTURE WORK

- Derive intervals for the noisy setting, as well as informed (but continuous) stochastic setting separately.

- With noisy samples using ECO's method it's theoretically possible for the reset ratio to become 'stuck' when the true gradient norm rapidly decreases. As the LR could be near zero, the LMS update would waste new samples until the bounds detect a new anomaly. Possible solutions may involve, just using Monte Carlo averaging when the noise is significant enough (and we don't want to smooth them per sample). Using a hybrid trust region function or alternative signals to reset.

- There may be stronger constraints or metric minimization's that can be placed on the Lagrange Definition of ECO's method, that for certain problems can achieve faster convergence.

- Formally define asymptotic bounds on $1/d \leq m \leq 1$ for specific problems, e.g. strong convexity constants, Lipschitz constants, non-smooth or non-convex. While the bounds of *SEGA* hold, we will consider if more can be proven.

- We haven't formalized the ratio methods to account for drift in the estimator, and the extended (or reduced) time to convergence that might add. The test is only for stationarity assumed, but a non-stationary factor would most likely be problem dependent. A possibility is to use a classic gradient trust region method to shrink and grow $\alpha$ or the confidence interval directly as a multiplier.

- We demonstrated $\mathcal{S}_{d-1}$ samples have provably faster convergence to the true gradient under Monte Carlo estimation for MSE minimizing, even if the difference is trivial. We suspect we'd get similar results deriving ECO's Method convergence under Gaussian samples. There may be other unbiased random distributions with provably faster convergence under these estimators with no more than $O(d \log(d))$ compute needs. e.g. a last-$n$ orthogonal RNG, which only needs to guarantee orthogonality with the last $n$ samples instead of all $d$.

- We can hypothetically use the expected angular bounds of $\hat{g}$ to $g$ accelerate true gradient convergence by sampling directional vectors in this range. This would be similar to an RL/policy gradient or even a modeled approach without knowing the action/state space. We would also investigate removing or altering the bias of this method.

- We hypothesize the $d$ independent T-distribution of low DOF significance levels; but it would be better to prove this. Or prove it's relation to another distribution.

# B PROOFS

**Lemma 2.1.** *Define $f(\boldsymbol{x})$ such that $\nabla f(\boldsymbol{x})$ is continuous, and let $\boldsymbol{u} \in \mathbb{R}^d$, s.t. $\|\boldsymbol{u}\| = 1$.*
*Then with $\theta = \angle(\nabla f, (v)\boldsymbol{u})$*

$$\frac{\|\nabla f(\boldsymbol{x}) - (v)\boldsymbol{u}\|}{\|\nabla f(\boldsymbol{x})\|} = \sin\theta, \qquad \frac{\|(v)\boldsymbol{u}\|}{\|\nabla f(\boldsymbol{x})\|} = \frac{|v|}{\|\nabla f(\boldsymbol{x})\|} = \cos\theta, \tag{2.1}$$

**Proof B.1** (*Lemma 2.1*)**.**
$\cos\theta$:

$$\cos\theta = \frac{((\nabla f(\mathbf{x}) \cdot \boldsymbol{u})\boldsymbol{u}) \cdot \nabla f(\mathbf{x})}{\|(\nabla f(\mathbf{x}) \cdot \boldsymbol{u})\boldsymbol{u}\|\|\nabla f(\mathbf{x})\|} = \frac{(\nabla f(\mathbf{x}) \cdot \boldsymbol{u})^2}{|\nabla f(\mathbf{x}) \cdot \boldsymbol{u}|\|\nabla f(\mathbf{x})\|} = \frac{|\nabla f(\mathbf{x}) \cdot \boldsymbol{u}|}{\|\nabla f(\mathbf{x})\|} = \frac{|v|}{\|\nabla f(\mathbf{x})\|}.$$

$\square$

$\sin\theta$:
Let $\nabla f(\boldsymbol{x}) = \boldsymbol{g}$

$$\|\boldsymbol{g} - (\boldsymbol{g} \cdot \boldsymbol{u})\boldsymbol{u}\|^2 = (\boldsymbol{g} - (\boldsymbol{g} \cdot \boldsymbol{u})\boldsymbol{u}) \cdot (\boldsymbol{g} - (\boldsymbol{g} \cdot \boldsymbol{u})\boldsymbol{u})$$
$$(\text{since } \boldsymbol{u} \cdot \boldsymbol{g} = \boldsymbol{g} \cdot \boldsymbol{u}, \ \boldsymbol{u} \cdot \boldsymbol{u} = 1)$$
$$= \|\boldsymbol{g}\|^2 - 2(\boldsymbol{g} \cdot \boldsymbol{u})^2 + (\boldsymbol{g} \cdot \boldsymbol{u})^2$$
$$= \|\boldsymbol{g}\|^2 - (\boldsymbol{g} \cdot \boldsymbol{u})^2.$$

Now:

$$\sqrt{\|\boldsymbol{g}\|^2 - (\boldsymbol{g} \cdot \boldsymbol{u})^2} = \|\boldsymbol{g}\|\sqrt{1 - \frac{(\boldsymbol{g} \cdot \boldsymbol{u})^2}{\|\boldsymbol{g}\|^2}}.$$

$$= \|\boldsymbol{g}\|\sqrt{1 - \cos^2\theta} = \|\boldsymbol{g}\|\sin\theta$$

And so:

$$\|\nabla f(\boldsymbol{x}) - (\nabla f(\boldsymbol{x}) \cdot \boldsymbol{u})\boldsymbol{u}\| = \|\nabla f(\mathbf{x})\|\sin\theta$$

$\square$

Under our definition of $\boldsymbol{u}$ we see that the gradient normalized *root mean square error* is $\sin\theta$, and $\sin\theta \le 1$ implies N-RMSE $\le 1$ and $0 \le \sin\theta$. Additionally we know that any real vector $\|\boldsymbol{v}\| \ge 0$ implies cosine is positive, so bounded by $0 \le \cos\theta \le 1$.

**Proof B.2** ((2.3) *Eco's Method*)**.**

By Lagrange

$$\mathcal{L}(\hat{\boldsymbol{g}}_k, \lambda) = \|\hat{\boldsymbol{g}}_k - \hat{\boldsymbol{g}}_{k-1}\|^2 + \lambda(\boldsymbol{u}^\top \hat{\boldsymbol{g}}_k - v).$$

$$\frac{\partial L}{\partial \hat{\boldsymbol{g}}_k} = 2(\hat{\boldsymbol{g}}_k - \hat{\boldsymbol{g}}_{k-1}) + \lambda\,\boldsymbol{u} = \boldsymbol{0}, \qquad \frac{\partial L}{\partial \lambda} = \boldsymbol{u}^\top \hat{\boldsymbol{g}}_k - v = 0$$

$$\hat{\boldsymbol{g}}_k = \hat{\boldsymbol{g}}_{k-1} - \tfrac{\lambda}{2}\boldsymbol{u}, \qquad\qquad \boldsymbol{u}^\top \hat{\boldsymbol{g}}_k = v$$

Then

$$v = \boldsymbol{u}^T \hat{\boldsymbol{g}}_{k-1} - \tfrac{\lambda}{2}\,\boldsymbol{u}^T \boldsymbol{u}$$

$$\lambda = \frac{2(\boldsymbol{u}^\top \hat{\boldsymbol{g}}_{k-1} - v)}{\boldsymbol{u}^\top \boldsymbol{u}}$$

$$\hat{\boldsymbol{g}}_k = \hat{\boldsymbol{g}}_{k-1} + \frac{(v - \hat{\boldsymbol{g}}_{k-1}^\top \boldsymbol{u})}{\boldsymbol{u}^\top \boldsymbol{u}}\,\boldsymbol{u}$$

By convex objective and affine constraint this is sufficient.

$\square$

**Proof B.3** (*Moment Contractions, MSE Shrinkage, Monte Carlo Convergence*).
For $\mathcal{N}(\mathbf{0}, \boldsymbol{I})$ we have moment generators (that old source):

$$\mathbb{E}\left[s_i s_j\right] = \delta_{ij}, \quad \mathbb{E}\left[s_i s_j s_k s_l\right] = \left(\delta_{ij}\delta_{kl} + \delta_{ik}\delta_{jl} + \delta_{il}\delta_{jk}\right).$$

For $\mathcal{S}_{d-1}$ we have moment generators (book source):

$$\mathbb{E}\left[s_i s_j\right] = \frac{\delta_{ij}}{d}, \quad \mathbb{E}\left[s_i s_j s_k s_l\right] = \frac{\delta_{ij}\delta_{kl} + \delta_{ik}\delta_{jl} + \delta_{il}\delta_{jk}}{d(d+2)}$$

*Some trivial proofs first*:

For $\mathcal{S}_{d-1}$ we get $\mathbb{E}[\boldsymbol{u}\boldsymbol{u}^T]\nabla f(\boldsymbol{x}) = \frac{1}{d}\nabla f(\boldsymbol{x})$

Which means for $\sqrt{d}\mathcal{S}_{d-1}$ we get $d \cdot \mathbb{E}[\boldsymbol{u}\boldsymbol{u}^T]\nabla f(\boldsymbol{x}) = \nabla f(\boldsymbol{x})$. Also:

$$\mathbb{E}[\cos\theta^2] = \mathbb{E}\left[\frac{(\nabla f(\boldsymbol{x}) \cdot \boldsymbol{u})^4}{|\nabla f(\boldsymbol{x}) \cdot \boldsymbol{u}|^2 \|\nabla f(\boldsymbol{x})\|^2}\right] = \frac{\nabla f(\boldsymbol{x})^T \mathbb{E}\left[\boldsymbol{u}\boldsymbol{u}^T\right]\nabla f(\boldsymbol{x})}{\|\nabla f(\boldsymbol{x})\|^2} = \frac{1}{d}$$

Now moving on, the law of total variance states:

$$\mathbb{E}\|\boldsymbol{X} - \mathbb{E}(\boldsymbol{X})\|^2 = \operatorname{tr}(\operatorname{Var}(\boldsymbol{X})).$$

$$\text{And } \operatorname{Var}(\boldsymbol{X}) = \mathbb{E}\left[\boldsymbol{X}\boldsymbol{X}^\top\right] - \mathbb{E}\left[\boldsymbol{X}\right]\mathbb{E}\left[\boldsymbol{X}\right]^T$$

Next we know that $\mathbb{E}[\boldsymbol{X}] = \boldsymbol{g}$ and associate $\boldsymbol{X} = \langle \boldsymbol{g}, \boldsymbol{u}\rangle\boldsymbol{u}$.

Then $\mathbb{E}\left[\boldsymbol{X}\boldsymbol{X}^\top\right]_{ij} = \sum_{p,q} g_p g_q \mathbb{E}\left[u_p u_q u_i u_j\right]$.

*Proof of Gaussian for Monte Carlo Estimator Equation* (2.2):

The Kronecker identity:

$$\mathbb{E}\left[\boldsymbol{X}\boldsymbol{X}^\top\right] = \|\boldsymbol{g}\|^2\boldsymbol{I} + 2\boldsymbol{g}\boldsymbol{g}^\top$$

$$\operatorname{tr}(\operatorname{Var}(\boldsymbol{X})) = \operatorname{tr}\left(\left(\|\boldsymbol{g}\|^2\boldsymbol{I} + 2\boldsymbol{g}\boldsymbol{g}^\top\right) - \boldsymbol{g}\boldsymbol{g}^\top\right) = (d+1)\|\boldsymbol{g}\|^2$$

And admits:

$$\text{N-MSE limit}: \ O(\frac{d+1}{k}), \ \text{N-MSE Adjustment}: \ \frac{k}{d+k+1}, \ \text{Adj. N-MSE limit}: \ O(\frac{d+1}{d+k+1}) \quad \text{(B.1)}$$

*Proof of $\mathcal{S}_{d-1}$ for Monte Carlo Estimator Equation* (2.2):

In this case, let us assume that $\boldsymbol{u} = \sqrt{d}\boldsymbol{s}$ so it matches the correct isotropic scaling for MC $\sqrt{d}\mathcal{S}_{d-1}$.

We instead get: $\mathbb{E}\left[u_i u_j\right] = \delta_{ij}, \quad \mathbb{E}\left[u_p u_q u_i u_j\right] = \frac{d}{d+2}\left(\delta_{pq}\delta_{ij} + \delta_{pi}\delta_{qj} + \delta_{pj}\delta_{qi}\right)$

And now scalar adjustment to previous result:

$$\mathbb{E}\left[\boldsymbol{X}\boldsymbol{X}^\top\right] = \frac{d}{d+2}\|\boldsymbol{g}\|^2\boldsymbol{I} + \frac{2d}{d+2}\boldsymbol{g}\boldsymbol{g}^\top$$

$2d/(d+2) - 1 = \frac{d-2}{d+2}$:

$$\operatorname{tr}(\operatorname{Var}(\boldsymbol{X})) = \operatorname{tr}(\frac{d}{d+2}\|r\|^2\boldsymbol{I} + \frac{d-2}{d+2}\boldsymbol{g}\boldsymbol{g}^\top) = \frac{d^2+d-2}{d+2}\|\boldsymbol{g}\|^2 = (d-1)\|\boldsymbol{g}\|^2$$

Admits:

$$\text{N-MSE limit}: \ O(\frac{d-1}{k}), \ \text{N-MSE Adjustment}: \ \frac{k}{d+k-1}, \ \text{Adj. N-MSE limit}: \ O(\frac{d-1}{d+k-1}) \quad \text{(B.2)}$$

$\square$

We see that Sphere Surface normalized directionals actually converge slightly quicker than basic gaussian, trivial at large dimension sizes, but valid at a small $d$.

**Proof B.4** (*ECO's Method Convergence Equation* (2.3)).
We can recognize ECO's Method as a form of randomized Kaczmarz update and refer to Gower and Richtarik's definition (see 3.3). We define it with $\boldsymbol{U}$, an arbitrarily finite set where every element $\|\boldsymbol{u}\| = 1$, and no $\boldsymbol{u}$ necessarily repeats. This is like

$$\boldsymbol{x}^{k+1} = \boldsymbol{x}^k - \frac{\boldsymbol{U}_{k:}\boldsymbol{x}^k - b_k}{\|\boldsymbol{U}_{k:}\|_2^2}(\boldsymbol{U}_{k:})^T$$

From Section 3.3 we find.

$$(3.4) \quad \mathbf{E}\left[\left\|\boldsymbol{x}^k - \boldsymbol{x}^*\right\|^2\right] \le \left(1 - \frac{\lambda_{\min}(\boldsymbol{A}^T\boldsymbol{A})}{\|\boldsymbol{A}\|_F^2}\right)^k \left\|\boldsymbol{x}^0 - \boldsymbol{x}^*\right\|^2$$

For our uniform isotropic sphere this reduces to:

$$\mathbf{E}\left[\|\hat{\boldsymbol{g}}_k - \nabla f(\boldsymbol{x})\|_2^2\right] = \left(1 - \min(\mathbb{E}[\boldsymbol{u}\boldsymbol{u}^T])\right)^k \|\nabla f(\boldsymbol{x})\|^2$$

Recall from B.3 the second moment of
$\mathcal{S}_{d-1}: \mathbb{E}\left[s_i s_j\right] = \frac{\delta_{ij}}{d}$. Which means that $\mathbb{E}[\boldsymbol{u}\boldsymbol{u}^T] = \frac{1}{d}\boldsymbol{I}$ and:

$$\mathbf{E}\left[\|\hat{\boldsymbol{g}}_k - \nabla f(\boldsymbol{x})\|_2^2\right] = \left(1 - d^{-1}\right)^k \|\nabla f(\boldsymbol{x})\|^2 \tag{B.3}$$

$\square$

**Proof B.5** (*True Directional Estimator Bounds corollary 3.1*).
Begin with a relation from lemma 2.1 and the definition of $c$ from 3.1 then:

$$\|\boldsymbol{g} - (v)\boldsymbol{u}\| = \sin\theta\|\boldsymbol{g}\|$$

$$\text{Generalized Equation (B.3)}: \quad \mathbf{E}\left[\|\hat{\boldsymbol{g}} - \boldsymbol{g}\|^2\right] = c^2 \|\boldsymbol{g}\|^2 \tag{B.4}$$

(If this is not self evident already) we know $\|\hat{\boldsymbol{g}}\|/\|\boldsymbol{g}\| = \cos\angle(\boldsymbol{g}, \hat{\boldsymbol{g}})$ so define $r = \|\hat{\boldsymbol{g}}\|$ and $\boldsymbol{m} = \hat{\boldsymbol{g}}/r$ so that $(r)\boldsymbol{m} = \hat{\boldsymbol{g}}$ (note a small difference is that $r$ will always be positive so $\boldsymbol{m}$ will always be on the right half of $\mathcal{S}_{d-1}$ but this shouldn't matter) now it follows from B.1 that $\hat{\boldsymbol{g}}$ satisfies lemma 2.1, and specifically $c = \sin\angle(\boldsymbol{g}, \hat{\boldsymbol{g}}) = \text{N-RMSE}[\hat{\boldsymbol{g}}, \boldsymbol{g}]$. Which let's us simplify:

$$\|\hat{\boldsymbol{g}} - \boldsymbol{g}\|^2 = c^2 \|\boldsymbol{g}\|^2, \quad \|\hat{\boldsymbol{g}} - \boldsymbol{g}\| = c \|\boldsymbol{g}\|$$

*Bounds ratio derivation:*
Our first attempt at stationary bounds involved solving the triangle inequality:
$\|\hat{\boldsymbol{g}} - \boldsymbol{g}\| + \|\boldsymbol{g} - (v)\boldsymbol{u}\| \le \|\boldsymbol{g}\|(\sin\theta + c_k)$.
But we can get stronger bounds:

$$\text{By Cauchy-Schwartz}: \quad |\langle \mathbf{u}, \mathbf{v}\rangle| \le \|\mathbf{u}\|\|\mathbf{v}\|, \ \|\mathbf{u}\| = 1.$$
$$\left\|\boldsymbol{u}^T\hat{\boldsymbol{g}} - \boldsymbol{u}^T\boldsymbol{g}\right\| = |\boldsymbol{u}^T\hat{\boldsymbol{g}} - v| \le \|\hat{\boldsymbol{g}} - \boldsymbol{g}\| = c\|\boldsymbol{g}\|$$
$$\frac{|\boldsymbol{u}^T\hat{\boldsymbol{g}} - v|}{\|\boldsymbol{g}\|c} \le 1$$

$\square$

**Proof B.6** (*ECO expectation ratio corollary 3.2*).
Where $u$ is our only random variable we get:

$$\begin{aligned}
\mathbb{E}\left[\left(\boldsymbol{u}^T\hat{\boldsymbol{g}} - \boldsymbol{u}^T\boldsymbol{g}\right)^2\right] &= \mathbb{E}\left[(\boldsymbol{u}^T(\hat{\boldsymbol{g}} - \boldsymbol{g}))^2\right] &&\triangleright \text{ let } \boldsymbol{y} = (\hat{\boldsymbol{g}} - \boldsymbol{g}) \\
&= \boldsymbol{y}^T\mathbb{E}\left[\boldsymbol{u}\boldsymbol{u}^T\right]\boldsymbol{y} &&\triangleright \text{ can't know } \boldsymbol{y}\boldsymbol{y}^T \\
&= \frac{\boldsymbol{y}^T\boldsymbol{y}}{d} &&\triangleright \text{ but } \|\boldsymbol{y}\| = c\|\boldsymbol{g}\| \\
&= \frac{c^2\|\boldsymbol{g}\|^2}{d}
\end{aligned}$$

However this merely provided us the expected value, we are interested in $\mathcal{L}\left[\left(\boldsymbol{u}^T\boldsymbol{g} - \boldsymbol{u}^T\hat{\boldsymbol{g}}\right)^2\right]$. Fortunately we know:

$$\lim_{d\to\infty} \text{Unif}(\mathcal{S}_{d-1}) \to \mathcal{N}(\boldsymbol{0}, \frac{1}{\sqrt{d}}\boldsymbol{I}), \quad \text{and vice versa.}$$

This is evident by noting that every $u \in \boldsymbol{u}$ from $\mathcal{N}(\boldsymbol{0}, \frac{1}{\sqrt{d}}\boldsymbol{I})$ is i.i.d. $\sigma = 1/\sqrt{d}$, then define our sample set as $d$ separate $u$'s. By the Central Limit Theorem as $d$ grows $\bar{u}^2 = \sum u^2/d = 1/d$ then $d \cdot \bar{u}^2 = \boldsymbol{u}^T\boldsymbol{u} = \|\boldsymbol{u}\| = 1$ which is in $\text{Unif}(\mathcal{S}_{d-1})$.

Now say:

$$\mathbb{E}_\alpha\left[|\boldsymbol{u}^T\hat{\boldsymbol{g}} - v|^2\right] = \frac{c^2\alpha^2\|\boldsymbol{g}\|^2}{d}$$

$\square$

Gaussian is convergent to $\text{Unif}(\mathcal{S}_{d-1})$ so we say that it's 'probably ok' to use gaussian error function at large enough $d$. But we welcome you to calculate the $\text{Unif}(\mathcal{S}_{d-1})$ error function if you would like. Also because $\boldsymbol{u}$ is our only random, independent of $g$ or $\hat{g}$ and $\alpha$ seems not to depend on the number of dimensions $d$, this provides more credence to the T - model independence.

**Proof B.7** (*Noisy Eco Ratio Equation* (3.4)).
In this framework we still expect $\|\hat{\boldsymbol{g}}_k - \boldsymbol{g}\| \approx \|\boldsymbol{g}\|c$ even if all our observations are noisy, but this is reasonable to estimate as we will see because it simply entails calculating the correct $c$.
We find that:

$$(\boldsymbol{u}^T\hat{\boldsymbol{g}} - v + e)^2 = (\boldsymbol{u}^T\hat{\boldsymbol{g}} - v)^2 + 2e(\boldsymbol{u}^T\hat{\boldsymbol{g}} - v) + e^2$$
$$\mathbb{E}[(\boldsymbol{u}^T\hat{\boldsymbol{g}} - \tilde{v})^2] = \mathbb{E}[(\boldsymbol{u}^T\hat{\boldsymbol{g}} - v)^2] + \sigma^2$$

So we get:

$$(\boldsymbol{u}^T\hat{\boldsymbol{g}} - \tilde{v})^2 \lesssim \frac{\alpha^2 c^2\|\boldsymbol{g}\|^2}{d} + b^2\sigma^2$$

And from here it's apparent how we get the noisy ratio. $\square$

**Proof B.8** (*Optimal Learning Rate for Noisy ECO's Method Equation* (3.5)).
Under the Lagrangian derivation of ECO's Method we know the optimal learning rate is $l = 1$ in the smooth setting. Instead of solving another constraint metric with noise, we recognize our method as a specific Normalized LMS setting and use it's system derived identities (source). (might need to change this lets see)

The optimal learning rate of N-LMS:

$$l_{\text{opt}} = \frac{E\left[|y(n) - \hat{y}(n)|^2\right]}{E\left[|e(n)|^2\right]}$$

Note the equivalences:

$$y(n) - \hat{y}(n) \Rightarrow \boldsymbol{u}^T\boldsymbol{g} - \boldsymbol{u}^T\hat{\boldsymbol{g}}_k = v - \boldsymbol{u}^T\hat{\boldsymbol{g}}_k.$$
$$e(n) = d(n) - \hat{y}(n) = y(n) + r(n) - \hat{y}(n)$$
$$\Rightarrow \boldsymbol{u}^T\tilde{\boldsymbol{g}} - \boldsymbol{u}^T\hat{\boldsymbol{g}}_k = v + e - \boldsymbol{u}^T\hat{\boldsymbol{g}}_k.$$

Now we have:

$$l = \frac{E\left[|v - \boldsymbol{u}^T\hat{\boldsymbol{g}}_k|^2\right]}{E\left[|v + e - \boldsymbol{u}^T\hat{\boldsymbol{g}}_k|^2\right]}$$

The first observation we can make is that $l \leq 1$ always, which is sensible as under perfect conditions $l = 1$.

From B.6 we have $\mathbb{E}\left[|v - \boldsymbol{u}^T\hat{\boldsymbol{g}}_k|^2\right] = \mu = \frac{c^2\|\boldsymbol{g}\|^2}{d}$.
From B.7 we get $\mathbb{E}\left[|v + e - \boldsymbol{u}^T\hat{\boldsymbol{g}}_k|^2\right] = \mu + \sigma_e^2$:

$$l_{\text{opt}} = \frac{\mu}{\mu + \sigma_e^2}$$

And we know $\hat{\mu}_k$ already from the section.

$\square$

## C PSEUDOCODES

The analytic solution to (3.3) for a partial reset, note that there can't be any noise term $\sigma$ and $\alpha$ is not from the T-model:

---

**Algorithm 1** Smooth Gradient Estimator Error Solve

---

**Require:** $u \in \mathbb{R}^n$, $\hat{g} \in \mathbb{R}^n$, $v \in \mathbb{R}$, $c \in [0, 1)$

1: **procedure**
2:      $c_n \leftarrow 1$                                         ▷ Initial value if no valid roots.
3:      $m \leftarrow \left(u^T \hat{g}\right)^2 + \alpha^2 \|\hat{g}\|^2 d^{-1} - v^2 \left(1 - c^2\right)$
4:      **if** $m > 0$ **then**
5:

$$\mathcal{C} \leftarrow \frac{\mp v\sqrt{1 - c^2}\frac{\alpha\|\hat{g}\|}{\sqrt{d}} \pm \left(u^T \hat{g}\right)\sqrt{m}}{\left(u^T \hat{g}\right)^2 + \alpha^2\|\hat{g}\|^2 d^{-1}}$$

6:

$$c_n = \min_{c'} c' \in [c_{++}, c_{+-}, c_{-+}, c_{--}],$$

$$\text{s.t. } c < c' < 1$$

7:      **end if**
8:      **return** $c_n$
9: **end procedure**

---

