# OpenReview forum: "ECO grad: Error Correcting Optimization for Quasi-Gradients, a Variable Metric DFO Strategy"
_ICLR.cc/2026/Conference — ICLR 2026 Conference Desk Rejected Submission_

### Official Review · Reviewer_zPed · 2025-10-23

**Soundness:** 1
**Presentation:** 1
**Contribution:** 1
**Rating:** 0
**Confidence:** 3

**Summary:**

This paper proposes a new algorithm for improving the  quasi-Newton method in reducing dimensional dependence, especially for the  DFO and no-grad problems.

**Strengths:**

This paper has nice figures. All references have links. Important formulas are boxed.

**Weaknesses:**

This submission is apparently incomplete. Many sections are completed empty, e.g. Section 1 and Section 3.4. In appendix, many drafts left unpolished. For this reason, it is hard to understand the backgroud and the main contribution of this work. Moreover, it is necessary to include more detailed information in each lemma theorem statements, especially in explaining or recapping the meaning of mathematical notations.

**Questions:**

As this submission is not complete, I do not have further questions in this work.

---

### Official Review · Reviewer_Wv5Y · 2025-10-27

**Soundness:** 3
**Presentation:** 1
**Contribution:** 1
**Rating:** 0
**Confidence:** 5

**Summary:**

The paper introduces a zero-th order quasi-Newton method, and a gradient estimator called ECO. However, the paper is incomplete and it is difficult to understand what is the real problem being studied and the proposed solution.

**Strengths:**

I cannot provide any positive feedback here, as the incompleteness of the paper makes it difficult to understand the content of the submission. I think the authors should complete the manuscript before submitting for publication.

**Weaknesses:**

1. The paper is incomplete. The first section is entirely empty, as is the results section. I cannot recommend anything other than an immediate reject for a paper with whole sections missing.

**Questions:**

None.

---

### Official Review · Reviewer_LsWh · 2025-11-02

**Soundness:** 1
**Presentation:** 1
**Contribution:** 1
**Rating:** 0
**Confidence:** 5

**Summary:**

This paper introduces ECO grad to estimate gradients using directional derivatives. The authors claim that this method achieves an exponential convergence rate in gradient MSE, which is significantly fast than Monte Carlo averaging.

**Strengths:**

The theoretical ideas of using LMS updates in gradient estimation and detecting true gradient change are interesting.

**Weaknesses:**

The paper is fundamentally incomplete and not in a state suitable for review. The submission appears to be a very early draft. The empirical validation of the ECO grad algorithm is missing.

**Questions:**

The manuscript must be completed with a full results section and all placeholders filled before it can be seriously evaluated by the community. For the current unfinished draft I don't have question.

---

### Official Review · Reviewer_vvuf · 2025-11-02

**Soundness:** 1
**Presentation:** 1
**Contribution:** 1
**Rating:** 0
**Confidence:** 5

**Summary:**

This paper is very difficult to read, or to summarize. There is no introduction section, the contributions aren't cleanly delineated, and the problem isnt' even clear. The paper seems to study the problem of estimating gradients given access only to sampling random distributions and directional derivative and gives a new algorithm "ECO's Method" (2.3) for doing that. The new method is sequential (and therefore we cannot do minibatching to estimate the gradient as in (2.2)), and is termed a "Quasi-Gradient method" as it "improves a gradient estimate at parameter $x_k$ that was already established at $x_{k-n}$ (which makes no sense, the gradients at different parameters can be significantly different unless you also control how far you move between them). The convergence proof (Proof B.2) is not really a proof, it just says this estimate minimizes some objective, but does not explain why minimizing this objective is meaningful at all, or what's the relationship to the true gradient. The authors claim exponential convergence of a “normalized RMSE” metric under unit sphere sampling via a Randomized Kaczmarz argument, but as I said I don't see how this argument follows at all when the parameters change. This is claimed as an advantage over Monte-Carlo averaging which yields sublinear rates (Appendices B.3-B.4). There is some discussion of convergence later with a certain "ECO ratio" introduced to detect estimator drift and trigger full/partial resets but again, I don't understand how this would still give a correct proof and provide the same rate. Section 3.4 is empty. There is very little in the way of empirical applications. This paper needs a significant amount of work, in the current shape it is just too difficult to assess.

**Strengths:**

1. Please see the summary section.

**Weaknesses:**

Please see the summary section. Below I summarize as points what I highlighted there.

1. There is no clear problem oracle. The text alternates between “zeroth‑order directional derivatives,” “forward gradients,” and “JVPs,” but never actually defines what oracle is assumed (function values with finite differences vs. direct access to $v=\langle\nabla f,u\rangle$ (how would you get that?!?) or its cost/noise model.
2. The method is described as improving a gradient estimate at $x_k$ using an estimator established at previous $x_{k-n}$. Without controlling ($\|x_k-x_{k-n}\|$) or the drift of $\nabla f$ (e.g., via Lipschitz‑based coupling of step sizes), I don't see how this works. The exponential‑rate claim uses randomized Kaczmarz with a *fixed* target vector, which does not account for changing $x_k$.
3. Incomplete or missing proofs. Proof B.2 only derives the update via a least‑change Lagrangian; it does not argue consistency, unbiasedness, or relation to the gradient.
4. The motivation for the ECO ratio is unclear. The ratio mixes a sphere‑sampling setup with Gaussian/t‑statistics approximations, it's all hand-wavy and doesn't really involve any proofs.
5. Even ignoring all the above, ECO’s Method is inherently sequential in $k$ whereas most applications use minibatching.
6. Inconsistent notation include “0TH‑ORDER,” inconsistent capitalization, no introduction section or clear statement of contributions. Here are two examples of statements in the manuscript that made no sense to me:
1. "Unbiasedness does not mean the distribution of u can’t be biased e.g. Rademacher or Bernoulli," What does this mean? Later in the same line we're directed to section A.1, which illucidates nothing more. The next section A.2 is incomplete.
2. "It's the only fully independent, identically distributed, and uniform variable on $\mathcal{S}_{d-1}$"?? That's the definition??

Is this some kind of experiment with an LLM-generated manuscript? If yes, that'd be such a flagrant violation of research ethics and it'd have taken up valuable peer-review time & effort that could be spent towards genuine research submissions.

**Questions:**

N/A.

---

### Note · Program_Chairs · 2026-01-17
**Submission Desk Rejected by Program Chairs**

The following references in this submission do not refer to real documents and/or have major errors in bibliographic information:

 Republic, Academy of Sciences of the Czech, ceka, Jan V1, sana;b, and Ladislav Luk$\cdot$. Institute of computer science. 201